# Management of Intraoperative Mechanical Ventilation to Prevent Postoperative Complications after General Anesthesia: A Narrative Review

**DOI:** 10.3390/jcm10122656

**Published:** 2021-06-16

**Authors:** Alberto Fogagnolo, Federica Montanaro, Lou’i Al-Husinat, Cecilia Turrini, Michela Rauseo, Lucia Mirabella, Riccardo Ragazzi, Irene Ottaviani, Gilda Cinnella, Carlo Alberto Volta, Savino Spadaro

**Affiliations:** 1Department of Translation Medicine and for Romagna, Section of Anesthesia and Intensive Care, University of Ferrara, 44121 Ferrara, Italy; federica.montanaro@edu.unife.it (F.M.); cecilia.turrini@hotmail.com (C.T.); rgc@unife.it (R.R.); irene.ottaviani@unife.it (I.O.); vlc@unife.it (C.A.V.); savinospadaro@gmail.com (S.S.); 2Department of Clinical Sciences, Faculty of Medicine, Yarmouk University, Irbid 21163, Jordan; Loui.husinat@yu.edu.jo; 3Department of Anesthesia and Intensive Care, University of Foggia, 71122 Foggia, Italy; michela.rauseo@hotmail.it (M.R.); lucia.mirabella@unifg.it (L.M.); gilda.cinnella@unifg.it (G.C.)

**Keywords:** general anesthesia, postoperative pulmonary atelectasis, respiratory failure, postoperative pulmonary complications, risk assessment, preoperative care, intraoperative monitoring, postoperative care, positive end expiratory pressure, precision medicine, intraoperative mechanical ventilation

## Abstract

Mechanical ventilation (MV) is still necessary in many surgical procedures; nonetheless, intraoperative MV is not free from harmful effects. Protective ventilation strategies, which include the combination of low tidal volume and adequate positive end expiratory pressure (PEEP) levels, are usually adopted to minimize the ventilation-induced lung injury and to avoid post-operative pulmonary complications (PPCs). Even so, volutrauma and atelectrauma may co-exist at different levels of tidal volume and PEEP, and therefore, the physiological response to the MV settings should be monitored in each patient. A personalized perioperative approach is gaining relevance in the field of intraoperative MV; in particular, many efforts have been made to individualize PEEP, giving more emphasis on physiological and functional status to the whole body. In this review, we summarized the latest findings about the optimization of PEEP and intraoperative MV in different surgical settings. Starting from a physiological point of view, we described how to approach the individualized MV and monitor the effects of MV on lung function.

## 1. Introduction

Mechanical ventilation (MV) is still necessary in many surgical procedures to provide gas exchanges during general anesthesia (GA) [1,2]. The concept of ventilator-induced lung injury has long been known; indeed, inadequate MV settings can lead to both atelectasis and lung overdistention [3,4,5]. Most studies on protective mechanical ventilation are focused on acute respiratory distress syndrome (ARDS) patients, where low tidal volume (VT) and an adequate positive end expiratory pressure (PEEP) are useful to minimize the dangerous effect of MV [6,7,8,9].

As described in ARDS patients, also during GA, higher tidal volume produces inflammatory reaction and pulmonary damages; as a result, many studies have found that the use of higher VT in patients undergoing GA increases morbidity and mortality [10]. On the opposite side, the use of intraoperative low tidal volume can reduce postoperative pulmonary complications (PPCs) [7].

In the last decades, research focused on development of protective ventilation strategies to prevent PPCs; indeed, MV should provide gas exchanges while minimizing lung stress and strain [3,11]. From the clinical point of view, this purpose can be reached by coupling a deep physiological understanding of the different ventilatory parameters and a continuous monitoring of their effects on the lungs. Several randomized clinical trials (RCTs) failed to find a specific ventilation strategy able to reduce PPCs [12,13,14]. Patients’ heterogeneity may be one of the main confounding factors leading to negative RCT. One size may not fit all, and the same MV settings may not be adequate for all patients. An individualized approach may contribute to overcome the challenge given by patient’s heterogeneity.

This narrative review aims to provide a current knowledge regarding how to set mechanical ventilation in different type of surgery (i.e., open abdominal surgery, laparoscopy and thoracic surgery) in order to reduce the risk of PPCs.

## 2. Materials and Methods

We performed a narrative review after a literature search in PubMed using the following Medical Subject Heading terms: general anesthesia, postoperative pulmonary atelectasis, respiratory failure, postoperative pulmonary complications, risk assessment, preoperative care, intraoperative monitoring, post-operative care, positive end expiratory pressure, precision medicine, intraoperative mechanical ventilation.

The selection criteria of this review are: (1) adult patients involved, (2) English paper, (3) papers regarding comparison between the most important methods of mechanical ventilation in different operating settings. Case reports and expert opinions were excluded.

## 3. The Role of Tidal Volume

The need of low intraoperative tidal volume (6–8 mL/kg of ideal body weight) during GA is raised by several observational studies [15], RCT [7], and meta-analysis [16,17]. We will try to clarify whether volutrauma occurs independently from all the other ventilatory settings.

Volutrauma is usually described as a dangerous mechanical ventilation effect characterized by lung overdistension [18]. The latter occurs when the energy applied distends the lung units repeatedly above its total lung capacity-associated strain [19], leading to inflammatory activation. Of note, in case of atelectasis, which is common during GA [20], portions of the lung can suffer from the total lung capacity-associated strain even during tidal volume ventilation [21]. This issue is linked to the concept of atelectrauma, where lung regions with different elasticity co-exist, and the junctions between these regions act as “stress risers” [22,23].

The occurrence of atelectasis is enhanced by the use of low VT ventilation; recently, a physiological study confirmed that the best PEEP (defined as the PEEP level associated with the lowest driving pressure) depends on tidal volume used, and lower tidal volumes usually require higher PEEP levels [24]. There are some clinical studies that confirm these physiological concepts. A large observational study investigating more than 29,000 patients identified the combination of low VT (6 mL/kg) and minimal PEEP (2–3 cmH_2_O) as a risk factor for postoperative mortality [15]. Accordingly, a meta-analysis of randomized trial showed that the use of low tidal volumes can reduce hospital length of stay only when PEEP was used [17]. Notably, a multicenter prospective study found a 12% increased risk of PPCs for milliliters per kilogram of PBW of VT, even when relatively safe range of VT are used (median 8 mL/Kg IBW) [9]. Hence, a VT equal to 8 mL/kg, even if widely considered protective, may be still too much for some patients.

On the other hand, some experimental studies showed that the dangerous effects of the VT are independent from other ventilator parameters. In animal models, high VT were associated with occurrence of VILI despite low plateau pressure and respiratory rate [25]; consequently, the impact of high VT on VILI was greater than those predictable by the analysis of the other ventilatory settings. [26]. Moreover, the damage lead by high VT was immediately detectable, whereas the damage effects given by other variables were slower [26]. It follows that high VT have per se the ability of generating volutrauma, regardless of other ventilatory settings.

## 4. The Role of PEEP

Whereas there is a wide consensus about adopting low VT during GA, how to set an adequate level of PEEP is more debated. In a widely known RCT, higher PEEP levels (12 cmH_2_O) were associated with higher hemodynamic impairment and no clinical benefits when compared to low PEEP (i.e., 2 cmH_2_O) [12]. Nonetheless, some authors advocated that this non-individualized approach failed to characterize patient population [27,28]. Following the concept that “one size does not fit all” and that physiological responsiveness should guide the inclusion into clinical trials [29], recently, many authors investigated the effect of setting the PEEP basing on physiological parameters.

Among the physiological parameters involved in PEEP setting, the driving pressure plays a pivotal role. Driving pressure (∆P) is the difference between plateau pressure and PEEP and can also be expressed as the ratio of tidal volume to respiratory system compliance (Vt/Crs). The strain is the measure of material deformation relative to its original state. During volume controlled MV, the change in lung volume is represented by VT and the initial lung volume, which corresponds to the functional residual capacity (FRC). Thus, global volumetric lung strain can be estimated as VT/FRC. Since the Crs correlates with the FRC [30], ∆P can be interpreted as an approximation of global lung strain [31].

An accurate measurement of respiratory mechanics would need an inspiratory hold, which is uncommon in anesthesia ventilators, but an acceptable approximation can be obtained by increasing end-expiratory pause to 30 or 40% [32,33]. Driving pressure provides an easily available surrogate of global lung strain. In a meta-analysis of individual patient data, high intraoperative ∆P was independently associated with PPCs [34]. Similar results were shown by an observational study on almost 70,000 patients with an increase in the odds ratio when ∆P > 12.5 cmH_2_O [8]. Finally, a randomized trial by Park et al. showed that a ∆P-guided ventilation (i.e., titrating PEEP on the lowest ∆P) was associated with a lower incidence of PPCs when compared with conventional ventilation in thoracic surgery [35]. However, we must underline that this randomized study was performed during one-lung ventilation, and therefore, its results could be not extendable to other settings. Furthermore, these results should be confirmed by a large multicenter RCT before being strong enough to generate a statement.

There are several factors which support the use of intraoperative ∆P to guide mechanical ventilation: wide available, easy to calculate and physiologically based. Notably, there are also some limitations. Firstly, most of the studies focused on ∆P were conducted in critically ill patients, and their results might not be suitable in patients with healthy lungs [36,37]. Moreover, the usefulness of ∆P is primarily related to its ability to estimate the lung stress and strain, but the ∆P is a measure of the mechanical characteristics of the whole respiratory system and not exclusively of the lungs [38]. Changes in chest wall compliance may therefore limit the usefulness of using ∆P as a safe limit to avoid PPCs. There are some common clinical situations in which chest wall compliance leads to a critical divergence between ∆P and transpulmonary pressures (pneumoperitoneum, intraabdominal hypertension, obesity, Trendelenburg position) which will be discussed further on this review. Furthermore, lung inhomogeneity can result in regional overdistension even at “safe” ∆P levels [21]. Therefore, particularly in high-risk patients, ∆P monitoring should not be enough to ensure a protective MV.

Finally, there is still no agreement regarding how to set PEEP according to ∆P. Indeed, ∆P may be constant over several PEEP levels. Whether this issue seems to not affect clinical outcomes in the context of ARDS [36], only few studies investigated this topic in OR settings. Expert’s opinion suggests minimizing both the ∆P and the plateau pressure; thus, when different PEEP levels are associated with the “best” ∆P achievable, the lowest one should be set [39]. Differently, in the ongoing “Designation” trial performed in open abdominal surgery, the highest PEEP associated with the lowest ∆P is set [40]. Another confounding factor is represented by the selection of PEEP with a decremental or incremental trial. The decremental trial consists of a recruitment maneuver (RM) followed by a step-down decreasing of PEEP (usually performed in step of 2 cmH_2_O); the incremental PEEP trial could not include a RM and is usually performed with a step-up increasing of PEEP level until a pre-specified safe limit. Recently, the physiologic effects of these two approaches have been compared in a randomized trial performed in thoracic surgery, which will be discussed in the dedicated sub-heading [41].

## 5. Recruitment Maneuvers

Due to the known association between GA and atelectasis, RMs are widely used to open the lung before PEEP application [42]. Indeed, from a physiological point of view, more pressure is required to open a collapsed alveolus in comparison to keep an alveolus open [43].

Recruitment maneuvers may be performed in several ways, and there is still no agreement on which one may carry the most favorable effects. A common approach requires the use of a CPAP application to reach a static pressure of 30–40 cmH_2_O for 30–40 s [7,44]; however, expert opinions suggest that graded rise of pressure could be better tolerated from a hemodynamic standpoint [45]. Therefore, more recent studies usually involve a stepwise RMs with incremental increase in PEEP [46,47,48] or tidal volume [49,50]. The target pressure is also variable, ranging from a plateau pressure of 30 cmH_2_O [49] to 45 [47] or a peak pressure of 50 cmH_2_O [51]. Several other differences can be described, including the duration of the RMs, the number of RMs during surgery, and the inspiratory:expiratory ratio used. Consequently, a recent meta-analysis investigating the effects of RM during GA described a dramatical inhomogeneity between studies [52].

Such inhomogeneity may partially explain why the effects of RMs on occurrence of PPCs is still unclear, particularly during laparotomy. Severgnini et al. [1] showed that RMs were able to reduce the occurrence of PPCs on the first postoperative day. Nonetheless, they compared a protective ventilation strategy with RM versus a non-protective ventilation without PEEP; thus, the real effect of RM is not inferable. Similarly, Weingarten et al. showed intraoperative benefits of RM when compared to a ZEEP strategy [53]. Notably, studies focused on RMs during GA do not take into account the recruitability of the lung, which may significantly differ in accordance with patient’s baseline comorbidities and surgical position [54]. In addition, RMs are usually well tolerated hemodynamically but seem to be not free from the risk of overdistention, as described also in critically ill patients [37]; a recent RCT, indeed, found higher levels of receptor for advanced glycation end-products, a marker of lung epithelial injury [55], in patients receiving intraoperative RMs [56]. Further studies are needed to elucidate which patients should receive intraoperative RM and how.

## 6. Mechanical Power

Due to the complexity of the interaction between the many respiratory variables, many efforts have been made to achieve a comprehensive analysis of the energy given by the ventilator to the patients. Mechanical power (MP) is a summary variable including all the components which can possibly cause VILI. The MP is calculated with the Formula:Power = RR × {〖VT^2^ × [½ × ELrs+RR × ((1 + I:E))/(60 × I:E) × Raw] + ∆〖VT^2^ × PEEP}
where RR is respiratory rate, VT is tidal volume, ELrs is respiratory system elastance, and Raw is airway resistance [57]. Higher values of MP have been associated lung injury; even so, studies performed in healthy lungs during general anesthesia are mostly conducted in animals [26,58,59].

The MP Formula can give to the anesthesiologist the ability to balance the effects of each respiratory parameter on the lungs. For example, the effect of tidal volume, which is squared in the Formula, is predominant. Further, it appears that the effect of PEEP is dichotomic: it increases the MP but also has the ability to reduce it through a reduction in ELrs. Finally, the MP Formula highlights that the respiratory rate, usually neglected when discussing the genesis of VILI, has a linear correlation with the amount of the energy delivered to the lungs.

Despite the robust physiological bases of MP, some limits should be considered. First, a validation of MP Formula in a large surgical population is still lacking, with only a small study performed in thoracic surgery [60]. Second, due to the complexity of the Formula, easier equations are being tested to allow an easier bedside calculation of MP [61,62]. Finally, despite low MP values, local damage is still possible in case of inhomogeneous ventilation with atelectasis and hyperinflation present at the same time.

## 7. Respiratory Rate

The role of respiratory rate in the genesis of VILI is usually underrated, with common guidelines advocating an intraoperative RR based on end-tidal carbon dioxide tension (ETCO_2_) rather than on mechanical proprieties [63]. While the primary role of RR could be to keep ETCO_2_ in range, we should also consider that RR is directly proportional to the mechanical power and thus is unlikely to be innocent in the genesis of VILI [64]. Pathophysiologically, a higher respiratory frequency results in higher flow rates to maintain a given VT [65]. Accordingly, in an experimental model, a reduction of respiratory rate improves indices of lung damage [66], and recently high ventilatory frequencies (≥14 breath/min) were associated with higher odds of PPCs in 102,632 patients [65].

Raising the minute ventilation with higher RR was common in the past to avoid hypercapnia, but now, there is a growing evidence that permissive hypercapnia may be even protective for the lung [67]. It follows that clinical indication for high RR during GA is particularly limited. Furthermore, when high respiratory rates are used, a continuous monitoring of flow–time curve is recommended due to the risk of developing intrinsic PEEP [63], which can be difficult to detect in operating room setting but can be responsible for dynamic pulmonary hyperinflation and hemodynamic consequences.

## 8. Expiratory Flow Limitation

Expiratory flow limitation (EFL) is a pathological condition characterized by a sharp reduction of expiratory flow associated with increased risk of PPCs in patients undergoing general anesthesia [68]. In mechanically ventilated patients, EFL is usually defined by the lack of increasing in the expiratory flow when PEEP is decreased, also called PEEP test [69]. During anesthesia, FRC values may shift below the closing capacity, causing collapsible small airways and, consequently, the “opening-closing” phenomena.

This can contribute to PPCs through different pathways. Such cyclic closure results in a reduction on expiratory flow together with a physical stress to the airway wall, which promote inflammation [70]. Moreover, EFL can cause an enhance of regional overdistention [69], which is difficulty detectable during GA. Furthermore, the occurrence of EFL during mechanical ventilation may impair the efficacy of postoperative cough and the clearance of secretions in smaller airways [71,72,73].

Given the relationship between occurrence of EFL and PPCs, a routinely assessment of EFL is suggested; this is particularly relevant because intraoperative EFL is often at least partially reversible. In a study involving ARDS patients, extrinsic PEEP was able to reduce intrinsic PEEP in EFL patients [74]. Accordingly, an observational study demonstrated a “paradoxical” response to PEEP in EFL patients, i.e., the decrease of hyperinflation when PEEP was increased [75]. This is probably due to the fact that application of PEEP may stabilize small airways and consequently improve lung emptying.

## 9. Fraction of Inspired Oxygen

Ironically, the last setting discussed in this review is probably the first ventilator parameter changed in clinical practice both during anesthesia induction [76] and during intraoperative hypoxemia. In a recent nationwide surgery conducted in Taiwan, indeed, high oxygen fraction (FIO2 > 0.8) was the common intervention adopted when oxygen saturation fell below 94% [77]. Even so, the adequate level of intraoperative FIO_2_ is usually neglected when discussing protective ventilation strategies.

From a pathophysiological point of view, there are pitfalls and advantages of using higher or lower FIO_2_. Higher FIO_2_ may promote higher tissue (muscle) oxygen saturation (StO_2_), particularly in the site of surgical incision; this enhances bacterial killing activity and may reduce surgical site infection (SSI) rate. Nonetheless, a large meta-analysis did not reach a definitive conclusion on the associated between higher and lower occurrence of surgical site infections [78]. Despite the lack of strong clinical evidence, the World Health Organization (Geneva, Switzerland) recommends the use of high FIO_2_ during surgery, as well as in the postoperative period, to prevent SSI [79].

On the other hand, high intraoperative FIO_2_ during GA may affect the respiratory system, leading to atelectasis formation [20] and bacterial growth [80], and the cardiovascular system [81], leading to decreased cardiac output and increased vascular resistance. Atelectasis is also one of the most commonly occurring pulmonary complication after GA and may affect postoperative recovery; indeed, atelectasis is detectable for as long as 4 days in the postoperative evaluation [82] and has been associated with occurrence of pneumonia [80]. In support of this hypothesis, two large retrospective studies showed that high intraoperative FIO_2_ was associated with a dose-dependent manner in major respiratory complications and with a 30-day mortality rate [83,84]. These results are not confirmed in an RCT comparing two “fixed” FIO_2_ rate (30% vs. 80%), where no benefit or harmful effects were showed [85].

## 10. Special Settings: Laparoscopy Surgery

Laparoscopic surgery is gaining increasing popularity due to the minor invasiveness of the procedure. It permits smaller incisions and reduces postoperative pain that may result in enhanced recovery after surgery [86]. However, laparoscopy procedures are not free from risks, in particular when discussing MV. The pneumoperitoneum, indeed, causes an augmented intrabdominal pressure (IAP) that critically affects respiratory mechanics [87].

Pneumoperitoneum in anesthetized patients determines a cephalad shift of the diaphragm, further reducing functional residual capacity (FRC) and promoting atelectasis formation [88]. Moreover, it determines a reduction in the respiratory system compliance by stiffening the chest wall component. Therefore, for the same tidal volume, a greater amount of pressure will be spent on the chest wall determining a reduction in lung distending pressures, which is the transpulmonary pressure [89].

The transpulmonary pressure (PL) is the real pressure applied to the lung parenchyma that determines the inflation, and it is derived by airway pressure minus pleural pressure (Paw − Ppl). During pneumoperitoneum, the decrease in PL may result in lung collapse, particularly in the dependent lung regions, worsening lung mechanics, and gas exchange, and favoring both PPCs and VILI [90]. Finally, the application of pneumoperitoneum resulted in an augmented intraoperative pulmonary shunt and can affect intraoperative oxygenation [91].

Several strategies have been proposed to reverse the respiratory mechanics modifications. PEEP may counterbalance the increased IAP and restore improved lung mechanics. In a prospective study, the PEEP needed during laparoscopy surgery to reduce shunt and improve Crs was higher than those in open abdominal surgery [91]. Patients undergoing laparoscopy needed a PEEP of 10 cmH_2_O to achieve the same physiological results showed in open abdominal surgery with a PEEP of 5 cmH_2_O [91].

Many laparoscopy procedures (urological, gynecological surgery) need the application of Trendelenburg position that further enhances the transmission of the IAP to the thorax. Cinnella et al. [92] demonstrated on gynecological patients undergoing laparoscopy surgery that pneumoperitoneum and Trendelenburg position worsen respiratory mechanics with an important derangement in chest wall component, ECW increased by 30% and EL increased by 20%. As a result, both the transpulmonary end-inspiratory and end-expiratory pressures decreased. They demonstrated that an open lung approach (OLA), consisting of a RM followed by the application of 5 cmH_2_O PEEP, induced alveolar recruitment, improved both ECW and EL and ultimately ameliorated gas exchange. RM may open the collapsed alveoli and improve arterial oxygenation. RM and PEEP after pneumoperitoneum induction improve respiratory system elastance and oxygenation both in healthy and obese patients [51,93].

Besides the physiological effects of PEEP and RM on the lungs, the real unresolved question stays in dosing this kind of interventions and tailoring them on the specific patient characteristics. Given that the role of ∆P in laparoscopy surgery, particularly during Trendelenburg position, is affected by the changing in chest wall compliance, an accurate monitoring of the effect of MV on the lungs is challenging. Recently, a new method has been introduced where the PEEP values are selected to counterbalance the detrimental effects of IAP; to achieve this aim, authors targeted PEEP values on individual IAP by applying 2 cmH_2_O of PEEP over IAP. Compared to standard PEEP, a PEEP value targeted on IAP (range 10–17 cmH_2_O) resulted in lower PL [94].

Another promising method to achieve the best compromise between lung collapse and overdistension is to set the PEEP using the electrical impedance tomography (EIT) monitoring. It allows a breath-to-breath assessment of dynamic change in lung volumes, and it has been applied in several fields including perioperative medicine [95]. During laparoscopy surgery, EIT may assess ventilation distribution homogeneity at different PEEP levels [96,97] EIT could also be used to optimize PEEP at bedside to maintain normal FRC and oxygenation [98]. Of note, a recent randomized trial showed that during laparoscopic radical prostatectomy an individualized PEEP (range 8–20) was associated with higher end-expiratory lung volumes and lower global inhomogeneity index (GI) when compared to 5 cmH_2_O of PEEP [99]. In this study, the PEEP was individualized according to the regional ventilation delay index, a parameter useful to estimate the amount of cyclic tidal recruitment [100]. Regional ventilation delay inhomogeneity (RVDI) is defined as the standard deviation of regional ventilation delay (RVD) in all pixels. RVD is a measure of temporal delay in the distribution of inspired air in different regions of the lung (i.e., the temporal heterogeneity occurring in the ventilated lung); given so, a smaller RVDI indicates a more homogeneous distribution. In addition, a small RCT showed that intraoperative EIT-guided PEEP setting can reduce also postoperative atelectasis [3].

Finally, setting “optimal” PEEP should improve arterial oxygenation without impairing hemodynamics [101,102,103,104]. Even if many studies have shown that increased PEEP levels have slight and transient effects on cardiac output [92], hemodynamic monitoring in high-risk patients is suggested.

## 11. Special Settings: Thoracic Surgery

During thoracic surgery, one lung ventilation (OLV) is used to facilitate surgical access. Arterial oxygenation is impaired during OLV due to the shunt through the nondependent lung [33]. Moreover, the occurrence of atelectasis in the ventilated lung further decreases oxygenation by reducing the aerated lung volume. The flow of the blood through not-aerated regions of the dependent lung enhances intraoperative shunt and impaired oxygenation. Besides the risk for intraoperative hypoxemia, patients undergoing thoracic surgery have also an increased risk of PPCs, because of preexisting disease processes, the loss of functional lung parenchyma (pulmonary resection), the surgical injury, and the detrimental effects of MV [105].

It is to underline that, in the context of OLV, the usually VT applied (i.e., 6–8 mL/kg IBW) would result in an unacceptable high VT in the ventilated lung, so that a further reduction in VT must be considered. A VT of 4–5 mL/kg IBW has recently been proposed [33,105,106]. The effect of low VT on patients undergoing OLV is associated with atelectasis formation, which is heightened by lateral decubitus and increases the pressure applied to the ventilated lung [105]. Historically, approaches to preventing intraoperative atelectasis during OLV endorsed the use of high VT [105], but the recent findings of the dangerous effects of high VT during OLV shifted the attention to the setting of an appropriate PEEP level. A randomized trial by Parks et al. emphasized the relevance of maintaining “safe” limit of ∆P also during OLV; they showed that a ∆P-guided ventilation (i.e., titrating PEEP on the lowest ∆P) was associated with a lower incidence of PPCs compared with conventional protective ventilation in thoracic surgery [35]. Even so, there is still no agreement on the correct PEEP setting in the OLV context.

In a recent study, Spadaro and coworkers [33] documented the physiological interplay between low VT and different PEEP levels (from ZEEP to 10 cmH_2_O) on oxygenation, respiratory mechanics and ventilation/perfusion mismatch during OLV. They showed that only a relatively high PEEP level (10 cmH_2_O) guarantees improvement of gas exchange, shunt reduction, and respiratory mechanics. In another study, Rauseo et al. [107] found that lower PEEP levels, i.e., 6 ± 0.8 cmH_2_O (range 5–8 cmH_2_O) were able to improve oxygenation and lung mechanics. The apparent discrepancy between the two groups could be explained by patient’s heterogeneity and by the different design of the two studies which includes the range of PEEP investigated and the use or not of RM.

After application of RM, PEEP can be titrated through a decremental trial, to optimize respiratory mechanics while minimizing alveolar over distension [106]. However, the effect of standard or individualized PEEP with or without RM during OLV on postoperative morbidity remains unclear. In a recent small RCT, an individualized PEEP strategy consisting in a stepwise increase in PEEP value was compared to a strategy which includes RM and individualized PEEP [41]. As a result, both strategies were able to decrease intraoperative shunt and to achieve a protective ventilation, as evaluated by intraoperative ∆P. Even so, sub-group analysis performed in patients with high baseline ∆P suggest some physiological benefit in RM group. Given that intraoperative RM are not free from risk of alveolar injury [56], one could argue that RM should be limited to patients with higher ∆P. Even so, recommendations based on strong clinical outcomes are still lacking.

## 12. Special Settings: Morbidity Obese Patients

With the increase in the global prevalence of obesity, there is a parallel rise in the proportion of obese patients referred for major surgery [108], which significantly affects anesthesiology practice. During intraoperative MV in obese patients, many pathophysiological alterations related to the underline disease should be considered.

The pathological distribution of the adipose tissue increases abdominal pressure, with a cephalic displacement of the diaphragm which produces a restrictive pattern; as a result, the loss in FRC during GA is greater when compared to non-obese patients [109]. The FRC reduction could imply an enhanced closing volume with consequent small airways closure, increase in atelectasis and intrapulmonary shunt, frequently associated with a reduction in lung compliance [110]. On the other hand, there is a significant increase in airway resistance which promotes expiratory flow limitation, [69,73] intra-tidal opening and closing phenomena, and a complete de-arrangement of V/Q mismatch; indeed, both low V/Q and high V/Q lung regions usually co-exist in this setting.

There is no evidence concerning the “best” ventilation in obese patients. It is to underline that even in this setting protective ventilation should be applied using low tidal volume (6–8 mL/Kg) calculated over IBW and an adequate PEEP levels. Even so, retrospective studies show that obese patients receive a non-protective TV more often than non-obese patients, and the risk of inappropriate TV rises with the severity of the disease [111].

PEEP titration could be challenging in obese patients during GA and should be personalized on patient’s respiratory mechanics and hemodynamic status. A large RCT, which did not consider the individual response to PEEP [13], compared low PEEP levels (4 cmH_2_O) versus higher PEEP level (12 cmH_2_O with RM). The results did not show significant differences in incidence of PPC, whereas higher rates of pleural pressure were identified in the higher PEEP group. Conversely, in a study comparing PEEP titration based on lung ultrasound (average PEEP 12 cmH_2_O) versus standard setting (PEEP 4 cmH_2_O), patients undergoing bariatric surgery had improvement in oxygenation and less incidence of PPC when randomized to “personalized” PEEP [112].

Another parameter that could be considered is the presence of airway opening pressure (AOP); in obese patients during laparoscopic surgery, Trendelenburg position and pneumoperitoneum increase AOP, and PEEP should be titrated also considering this pressure and its changes over time during the surgery [113]. Indeed, the dynamic changes in respiratory mechanics associated with surgical position are magnified in obese patients. Tharp et al. [54] performed a study in 91 obese patients during laparoscopic surgery and collected respiratory system mechanical characteristics using esophageal pressure; patients were in Trendelenburg position of almost 30 degrees. Transpulmonary ∆P was higher in more obese patients in all the phases of surgery; also, end-expiratory transpulmonary pressure (i.e., the difference between end expiratory airway pressure and end expiratory esophageal pressure) became negative in Trendelenburg position in more obese patients, as mechanical mechanism for atelectasis. Accordingly, the optimal PEEP (i.e., a PEEP value equal to end-expiratory transpulmonary pressure) increased with pneumoperitoneum and further with Trendelenburg and ranged from 0 to 37 cmH_2_O.

The results by Tharp et al. gain relevance due to their measurement of esophageal pressure, which allowed an estimation of transpulmonary ∆P instead of using “simple” respiratory system ∆P. [54] The role of ∆P in the context of obese patients, indeed, deserves deeper analysis. Obese patients are characterized by higher elastance of the chest wall; thus, the divergence between ∆P (an indirect measure of whole respiratory system compliance) and transpulmonary pressure may significantly raise in this setting [114,115]; due to this discrepancy, the respiratory system ∆P in obese patients seems to not be associated with lung injury [114], whereas it may represent a parameter to estimate the negative effect of ventilation on right ventricular function [116]. Notably, data regarding ∆P in obese patients are mostly derived from critically ill setting, and further studies are needed to describe its role in OR settings.

In addition to the esophageal catheter there are other useful tools to guide the ventilatory setting, which is also to describe the regional differences in ventilation distribution. The EIT could be used to assess regional distribution of ventilation and to guide appropriate titration [117]. A study performed in 37 patients undergoing laparoscopic bariatric surgery used the EIT to evaluate the need for PEEP adjusting during pneumoperitoneum; EIT measurements confirmed an increasing PEEP demand during pneumoperitoneum [118].

Regardless the PEEP titration, the use of RM in obese patients has been widely studied in obese patients to reopen collapsed areas of the lung and allow a more homogeneous distribution of ventilation [119,120,121]. These data are confirmed by several studies on intraoperative RM, resulting in improved lung and respiratory system mechanics and oxygenation [122]. A RM performed after endotracheal intubation (consisted of applying a CPAP 40 cmH_2_O for 40 s) was able to improve gas exchange and end-expiratory lung volume when compared to PEEP alone [93]. Similar results were shown on patients undergoing sleeve gastrectomy [120] or gastric bypass [123]. Notably, the effect of RM seems to be limited to intraoperative period; indeed, in a randomized study by Defresne et al. RMs did not improve postoperative lung function including FRC, arterial oxygenation, and the incidence of obstructive apnea [124]. Similar results were shown by Whalen et al. [51]. It seems that a correct perioperative management of obese patients must consider postoperative period as well as intraoperative MV.

Finally, little is known about one-lung ventilation in obese patients undergoing thoracic surgery, but high levels of PEEP associated with RM seem to prevent atelectasis of the ventilated lung and to reopen closed alveoli, thus improving oxygenation and preventing postoperative hypoxemia [125]. Table 1 resumes the different strategies used to individualized PEEP described.

## 13. From Protective to Personalized: The Future of Intraoperative Mechanical Ventilation

The continuous growing of monitoring tools available at bedside is challenging the actual concept of protective ventilation. As stated before, the same ventilatory setting can be or not be “protective” depending on the physiological variables of the patients. As example, a “conventional” ventilation (i.e., low tidal volume and PEEP = 5 cmH_2_O) resulted in a non-homogeneous ventilation during robot-assisted laparoscopic prostatectomy with pneumoperitoneum and a steep Trendelenburg position, as assessed with EIT [126]. EIT can give additional information to those given by respiratory mechanics. Respiratory mechanics can better asses the dynamic stress, whether EIT may help to optimize lung recruitment and homogeneity of ventilation [127]. RVDI and GI can carry evidence regarding the temporal heterogeneity occurring in the lungs which are not available without EIT monitoring [100,128]. Moreover, data regarding regional air trapping are gaining importance in EIT evaluation and may represent an important adding to intraoperative MV knowledge [71]. Future studies should assess how this information can change our MV strategies and how to commute this advance monitoring ability in better clinical outcomes.

The same concept (i.e., coupling monitoring ability with clinical intervention) can be extended to the usefulness of intraoperative lung ultrasound assessment. Perioperative lung ultrasound has been used to dynamically detect the development of intraoperative atelectasis [129] or alveolar consolidation [130] as well as postoperative diaphragm dysfunction [131]. Given that lung ultrasound can assess PEEP-induced lung recruitment [132,133,134], its application could help to identify which patients could benefit from higher PEEP or recruiting maneuvers.

As resumed in this review, setting an adequate “personalized” MV able to optimize the lung function is far from being simple. Identifying the optimal MV strategy when considering the whole organism, and not only the lung, is even more challenging. Mechanical ventilation can affect the hemodynamic status of the patients in several ways, particularly with PEEP titration [133]. Briefly, the same PEEP value able to optimize lung function can impair cardiac output while resulting in lower arterial oxygen delivery (DO_2_) despite higher alveolar oxygen content; only few studies investigated the effects of different PEEP values on lung protection and DO_2_, showing that in a consistent percentage of patients, incremental PEEP appears to protect alveoli but resulted in lower DO_2_ [134].

The different systemic consequences of PEEP underline that the ventilator-induced lung injury is only one of the putative adverse effect of MV; recently, it has been shown that two MV strategies with same lung-protection ability can affect in different ways the cardiovascular system [26]. How much is the acceptable fall in DO_2_, and how to balance the lung and hemodynamics effects of MV, are far from being demonstrated.

Finally, it is worth underlining that the microcirculatory effects of MV are not fully explainable with changes in cardiac output. For example, PEEP application can affect renal blood flow with a non-linear relationship difficult to predict [135]. Therefore, specific organ monitoring is recommended particularly in the high-risk setting; recently, intraoperative Doppler-determined renal resistive index (RRI) has been identified as a risk factor for postoperative acute kidney injury in patients undergoing cardiopulmonary bypass [136].

## 14. Conclusions

In conclusion, the search for a holy grail in the setting of intraoperative mechanical ventilation is a false myth. Starting from the “low VT” dogma, several different settings should be applied depending on patient’s responsiveness and surgical setting. Given the multimodal effects of MV on the whole body, trying to reduce its complexity would result in a loss of characterization of the patients. The future success of different MV strategies will depend on their ability to improve specific patient population outcomes.

## Figures and Tables

**Table 1 jcm-10-02656-t001:** Physiological parameters used to individualize positive end-expiratory pressure.

Parameter Evaluated	Surgical Setting	How to Individualize PEEP	Range of PEEP Studied	Monitoring Tool	Results
∆P [8,34,35,39,40]	Open abdominal surgery [8,34,39,40] Laparoscopy [34,38] Thoracic surgery [34,35]	The lowest PEEP associated with the lowest ∆P [35,39] The highest PEEP associated with the lowest ∆P [38,40]	N/A	None	∆P < 13 cmH_2_O could reduce PPCs Targeting low ∆P results in low ∆P_L_
EFL [67]	Open abdominal surgery	PEEP value able to reverse EFL	N/A	None	Lower PPCs in patients with reversed EFL
Pulmonary shunt [91]	Open abdominal surgery Laparoscopy	PEEP associated with the lowest shunt	0 to 10 cmH_2_O	Beacon system	Better oxygenation
Intra-abdominal pressure [94]	Laparoscopy	PEEP = IAP + 2 cmH_2_O	10 to 17 cmH_2_O	IAP measurement	Lower ∆P_L_
RVDI [95]	Laparoscopy	PEEP associated with the lowest RVDI	8 to 20 cmH_2_O (IQR)	EIT	Better oxygenation, higher EELV
FRC [98]	Laparoscopy	PEEP able to maintain stable FRC	0 to 20 cmH_2_O	EIT	Normal FRC, low shunt
Lung collapse and hyperdistension [3]	Laparoscopy	Best compromise between collapse and hyperdistension	6 to 16 cmH_2_O	EIT	Lower postoperative atelectasis, lower intraoperative ∆P
Airway closure [113]	Laparoscopy, obese patients	PEEP value able to reach AOP	5 to 10 cmH_2_O	None	High rate of airway closure with PEEP range studied
P_L_ [118]	Laparoscopy, obese patients	PEEP value able to reach P_L_ = 0 cmH_2_O	10 to 25 cmH_2_O	Esophageal catheter	Positive P_L_
∆P_L_ [54]	Laparoscopy, obese patients	PEEP associated with lowest ∆P_L_	0 to 37 cmH_2_O	Esophageal catheter	Lower ∆P_L_

PEEP: positive end-expiratory pressure; ∆P: driving pressure; EFL: expiratory flow limitation; RVDI: regional ventilation delay index; FRC: functional residual capacity; P_L_: transpulmonary pressure; ∆P_L_: transpulmonary driving pressure.

## Data Availability

Not applicable.

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
