# Peer review of "Management of Intraoperative Mechanical Ventilation to Prevent Postoperative Complications after General Anesthesia: A Narrative Review"

_jcm, 2021, doi:10.3390/jcm10122656_

Round 1

Reviewer 1 Report

After extensive revision and inclusion of further literature of high relevance, the quality of the manuscript has been clearly improved. Above all, the indication that his individualized ventilation is the right approach can only be underlined. However, there are still some issues to be addressed.

However, the basic problem and a main point of criticism remains unaffected, which leads to the fact that in this form the manuscript cannot be accepted. Furthermore, it contains suggestions in the form of flow charts on how an individual PEEP can be set, without these being evaluated in any way. These are not validated by clinical studies, are not supported by current literature  and thus should not be presented in a review as recommendations. Therefore, I urgently recommend deleting these suggestions on how to set individual PEEP in different settings.

General comments:

  • The whole manuscript needs language editing
  • The paragraph dealing with mechanical power should be moved to an earlier section of the manuscript, as mechanical power is already mentioned in line 89-92 and in Figure 1 without being explained before. This might be hard to follow for the non-expert reader.

Specific comments:

Abstract:

  • Since the manuscript is now more focused on the individualization of PEEP, this should also be clarified in the abstract.

 Materials and Methods 

  • In the „Response to Reviewers“, you state that additional key words for the literature research were included. However, they are still not mentioned in the Materials and Methods section.
  •  

The role of tidal volume

  • I think the study of Park et al (Ref 35) should not be discussed here as thoracic surgery requiring one-lung ventilation is a different setting. Mixing studies with conventional ventilation and one-lung ventilation is confusing for the reader. At least, it should be clearly emphasized that the study of Park et al. refers to one-lung ventilation.

Figure 1: “EFL” should be explained in the Figure legends

How to set PEEP

  • Park et al Anesth 2019 study was not sufficiently powered to make a general statement regarding the incidence of PPC (at best with single-lung ventilation), although it has a signal in that direction. Therefore, this study is not sufficient evidence that individual PEEP determination based on driving pressure positively affects clinical outcome. This needs to be discussed more clearly.
  • Please delete figure 1 (also 2 and 3) if there is insufficient evidence. Please keep in mind that reviews are read by many and suggested methods may be uncritically adopted in clinical practice. Therefore, these should be justified by sufficient evidence.

Respiratory rate

Line 224: “…was common in the past to avoid hypocapnia” should be “hypercapnia”.

 Expiratory flow limitation

  • I do not understand the link of mechanical ventilation and the limitation of efficacy of cough as coughing should be generally avoided during general anesthesia and the studies citated refer to different settings than general anesthesia. Do you mean postoperative impairment of coughing? Please clarify.
  • Line 245: I guess you mean “extrinsic PEEP” rather than “external PEEP”

Special settings: laparoscopy surgery

Figure 2: I think that introducing EIT as advanced monitoring after beginning of the surgical procedure is not an option in the OR. Placing the belt intraoperatively is at least difficult – if not impossible, for example if special positioning devices (such as a vacuum mattress) are used for laparoscopic surgery in steep Trendelenburg position. Advanced monitoring such as EIT has therefore to be considered preoperatively if difficulties during MV are expected.

Special settings: Thoracic surgery

 Figure 3: Why start with an FiO2 < 60% for OLV? For safety reasons, I would always start with an FiO2 > 60% (even > 80%) at the initiation of OLV and then titrate FiO2 by repeated arterial blood gas analysis. Furthermore, I would first raise FiO2 to 100% before starting CPAP in the nonventilated lung. However, raising FiO2 is not listed as a potential option in the flow chart. Please explain why.

Mechanical ventilation in obese patients

  • Line 447-450: EIT is not suitable to assess transpulmonary pressure because it does not allow pressure measurements.

From protective to personalized: the future of intraoperative mechanical ventilation

  • Mechanical ventilation with a PEEP of 5 cmH2O should not be called protective ventilation.

Reviewer 2 Report

This is an excellent, readable, erudite narrative review. In my opinion, this review is more useful to the average reader than the technically complex, verbose, and complex metanalyses and structured reviews (while not replacing them, of course).

The writing is cursive, clear, and easy to follow, the structure is simple yet comprehensive.

The only deficit is that the review addresses a large number of topics, and in just a few of them does it attempt to present a balanced representation of the complexity of the science relevant to the topic. This is of course also the strength of the review, but I would expect the authors to stress this issue in a "Limitations" paragraph at the end.

Author Response

This manuscript is a resubmission of an earlier submission. The following is a list of the peer review reports and author responses from that submission.

Round 1

Reviewer 1 Report

Thank you for the opportunity to review this manuscript. Intraoperative mechanical ventilation can increase the risk of postoperative pulmonary complications and it is still not clear how to reduce the risk by optimising intraoperative ventilation. Positive end-expiratory pressure seems to play a special role. The "one size fits all" approach and the use of a higher, identical PEEP for all patients without taking individual circumstances into account does not seem to lead to a reduction in PPC (ProVHILO and PROBESE trials). Therefore, the question arises whether an individualisation of ventilation could perhaps reduce the risk of PPC. However, this question has not yet been conclusively clarified.

General comments:

The authors of this review attempt to provide an overview of the current state of research and derive a new pathway for individualising PEEP during general anaesthesia and for two special cases (laparoscopy and one-lung ventilation). However, this review is not suitable for this purpose in this form.

The main weaknesses of the review lie in the literature search, which does not provide a comprehensive picture of the current literature. In addition, the title suggests an individualisation of the entire mechanical ventilation, but in fact "only" the PEEP is individualised. However, the review does not present different methods for PEEP titration, but only the driving pressure, which makes a misinterpretation possible and represents a very one-sided view. However, the use of driving pressure to optimise mechanical ventilation has not been conclusively clarified and cannot be generally recommended. Further clinical studies are still necessary. In addition, special patient groups are unfortunately not examined, which is imperative when writing a review on the individualisation of mechanical ventilation.

Furthermore, the authors repeatingly mix studies concerning PEEP selection in ARDS patients and the ICU, but the intention of the manuscript is to review best PEEP selection in the operating room. This may be due to the fact that literature on best PEEP selection in the OR is scarce, but this may confuse the reader. Furthermore, concepts of best PEEP selection in the ICU might not be suitable in the OR in patients with healthy lungs.

Due to these points of criticism, a comprehensive revision with further literature research and revision of the pathways as well as complete further chapters is necessary.

Specific comments:

Major:

  1. The term „a narrative review” should be included in the title to make clear that the authors did not perform a systemativ review of the current literature (see below).
  2. After critical reading and consideration of the proposed pathways, the authors discuss various ventilation parameters, but in the end only the individualisation of PEEP is proposed in the pathway. Therefore, the title should rather read: "Management of individual PEEP during mechanical ventilation to prevent postoperative complications after general anesthesia".
  3. I find the selection of keywords for the literature search unsuccessful. Essential keywords such as intraoperative ventilation, individualisation, PEEP, PPC are missing, but are absolutely necessary to get a more comprehensive picture of the current literature. The literature search should be carried out again with additional keywords.
  4. Although individualisation of ventilation (PEEP) is mentioned, it is not made clear why. It should be discussed to what extent global ventilation parameters are suitable for individualisation at all, or whether regional parameters would be more suitable and with which technique these can be collected. Further possibilities of setting PEEP by means of compliance, etc., but also the use of EIT and its parameters such as GI and RVDi should be discussed and their advantages and disadvantages presented.

Girrbach F, Petroff D, Schulz S, Hempel G, Lange M, Klotz C, Scherz S, Giannella-Neto A, Beda A, Jardim-Neto A, Stolzenburg JU, Reske AW, Wrigge H, Simon P: Individualised positive end-expiratory pressure guided by electrical impedance tomography for robot-assisted laparoscopic radical prostatectomy: a prospective, randomised controlled clinical trial. Br J Anaesth. 2020 Sep;125(3):373-382

  1. Driving pressure as a global parameter has not yet been shown in any study to be suitable for setting PEEP. However, it is now recommended as a parameter in this review. This has to be discussed critically and the authors should explain it. What are the proposed pathways based on? Have they been evaluated in studies? Only then should they be recommended. I find it very critical to formulate such recommendations without prior clinical testing, especially since the current literature does not provide sufficient data to allow such an interpretation.

Writing Group for the Alveolar Recruitment for Acute Respiratory Distress Syndrome Trial (ART) Investigators, Cavalcanti AB, Suzumura ÉA, Laranjeira LN, Paisani DM, Damiani LP, Guimarães HP, Romano ER, Regenga MM, Taniguchi LNT, Teixeira C, Pinheiro de Oliveira R, Machado FR, Diaz-Quijano FA, Filho MSA, Maia IS, Caser EB, Filho WO, Borges MC, Martins PA, Matsui M, Ospina-Tascón GA, Giancursi TS, Giraldo-Ramirez ND, Vieira SRR, Assef MDGPL, Hasan MS, Szczeklik W, Rios F, Amato MBP, Berwanger O, Ribeiro de Carvalho CR: Effect of Lung Recruitment and Titrated Positive End-Expiratory Pressure (PEEP) vs Low PEEP on Mortality in Patients With Acute Respiratory Distress Syndrome: A Randomized Clinical Trial. JAMA 2017; 318(14):1335-1345

Amato MB, Meade MO, Slutsky AS, Brochard L, Costa EL, Schoenfeld DA, Stewart TE, Briel M, Talmor D, Mercat A, Richard JC, Carvalho CR, Brower RG: Driving pressure and survival in the acute respiratory distress syndrome. N Engl J Med 2015; 372(8):747-55

  1. How exactly should PEEP be determined after driving pressure? How should PEEP be titrated after driving pressure? Should a decremental PEEP trial be performed? When performing a decremental PEEP trial, different strategies exist for the choice of the best PEEP value according to driving pressure, because the driving pressure may be constant over several PEEP steps. For example, in the ongoing DESIGNATION trial, the highest PEEP is chosen in this case, while several authors advocate to choose the lowest PEEP. The detailed pathway to select the best PEEP is not clear enough in the review and the pathways. In this context, the use of recruitment manoeuvres also needs to be discussed.
  2. In general, the use of recruitment manoeuvres and their objective should be discussed in more detail and a recommendation formulated. In general? In the context of PEEP titration?

Rothen HU, Neumann P, Berglund JE, Valtysson J, Magnusson A, Hedenstierna G: Dynamics of re-expansion of atelectasis during general anaesthesia. Br J Anaesth 1999; 82:551–6

 Imber DA, Pirrone M, Zhang C, Fisher DF, Kacmarek RM, Berra L: Respiratory management of perioperative obese patients. Respir Care 2016; 61(12):1681-1692

 Slutsky AS, Ranieri VM: Ventilator-induced lung injury [published correction appears in N Engl J Med 2014; 370(17):1668-9]. N Engl J Med 2013; 369(22):2126-2136

 Defresne AA, Hans GA, Goffin PJ, Bindelle SP, Amabili PJ, DeRoover AM, Poirrier R, Brichant JF, Joris JL: Recruitment of lung volume during surgery neither affects the postoperative spirometry nor the risk of hypoxaemia after laparoscopic gastric bypass in morbidly obese patients: a randomized controlled study. Br J Anaesth 2014; 113(3):501–7

Writing Group for the Alveolar Recruitment for Acute Respiratory Distress Syndrome Trial (ART) Investigators, Cavalcanti AB, Suzumura ÉA, Laranjeira LN, Paisani DM, Damiani LP, Guimarães HP, Romano ER, Regenga MM, Taniguchi LNT, Teixeira C, Pinheiro de Oliveira R, Machado FR, Diaz-Quijano FA, Filho MSA, Maia IS, Caser EB, Filho WO, Borges MC, Martins PA, Matsui M, Ospina-Tascón GA, Giancursi TS, Giraldo-Ramirez ND, Vieira SRR, Assef MDGPL, Hasan MS, Szczeklik W, Rios F, Amato MBP, Berwanger O, Ribeiro de Carvalho CR: Effect of Lung Recruitment and Titrated Positive End-Expiratory Pressure (PEEP) vs Low PEEP on Mortality in Patients With Acute Respiratory Distress Syndrome: A Randomized Clinical Trial. JAMA 2017; 318(14):1335-1345

  1. The determination of mechanical power is an interesting topic, but overestimated in the current presentation in the review. So far, it has not been reviewed in any major study. In my view, this paragraph should be presented much more openly and can also be shortened to make room for other discussions such as different methods for individualising the PEEP.
  2. In other studies on perioperative individualised PEEP settings, PEEP values of more than 15 cmH2O are sometimes determined and are significantly higher than the PEEP values proposed here (Nestler et al. BJA 2017, Girrbach et al. BJA 2020). Why do the authors propose such low PEEP values?
  3. Specific patient groups are addressed but not discussed further. Especially in obese patients, numerous studies have shown that they require an adjustment of the mechanical ventilation and are likely to benefit from individualisation. The pathways recommended by the authors will not work in obese patients, even from their own experience, and are not recommended for this patient group. This is a major weakness of the review and needs to be revised and supplemented. Further literature suggestions from me would be:

Reinius H, Jonsson L, Gustafsson S, Sundbom M, Duvernoy O, Pelosi P, Hedenstierna G, Fredén F: Prevention of atelectasis in morbidly obese patients during general anaesthesia and paralysis: a computerized tomography study. Anesthesiology 2009; 111:979–87

Nestler C, Simon P, Petroff D, Hammermüller S, Kamrath D, Wolf S, Dietrich A, Camilo LM, Beda A, Carvalho AR, Giannella-Neto A, Reske AW, Wrigge H: Individualized positive end-expratory pressure in obese patients during general anaesthesia. Br J Anaesth 2017; 119(6):1194-1205

Ball L, Hemmes SNT, Serpa Neto A, Bluth T, Canet J, Hiesmayr M, Hollmann MW, Mills GH, Vidal Melo MF, Putensen C, Schmid W, Severgnini P, Wrigge H, Gama de Abreu M, Schultz MJ, Pelosi P, Las Vegas Investigators; PROVE Network; Clinical Trial Network of the European Society of Anaesthesiology: Intraoperative ventilation settings and their associations with postoperative pulmonary complications in obese patients. Br J Anaesth 2018; 121(4):899-908

Imber DA, Pirrone M, Zhang C, Fisher DF, Kacmarek RM, Berra L: Respiratory management of perioperative obese patients. Respir Care 2016; 61(12):1681-1692

  1. A selection of additional literature that should be considered:

Las Vegas Investigators: Epidemiology, practice of ventilation and outcome for patients at increased risk of postoperative pulmonary complications: Las Vegas—an observational study in 29 countries. Eur J Anaesthesiol 2017; 34(8):492-507

Fernandez-Bustamante A, Frendl G, Sprung J, Kor DJ, Subramaniam B, Martinez Ruiz R, Lee JW, Henderson WG, Moss A, Mehdiratta N, Colwell MM, Bartels K, Kolodzie K, Giquel J, Vidal Melo MF: Postoperative pulmonary complications, early mortality, and hospital stay following noncardiothoracic surgery: a multicenter study by the Perioperative Research Network Investigators. JAMA Surg 2017; 152(2):157-166

Karalapillai D, Weinberg L, Peyton P, Ellard L, Hu R, Pearce B, Tan CO, Story D, O'Donnell M, Hamilton P, Oughton C, Galtieri J, Wilson A, Serpa Neto A, Eastwood G, Bellomo R, Jones DA: Effect of Intraoperative Low Tidal Volume vs Conventional Tidal Volume on Postoperative Pulmonary Complications in Patients Undergoing Major Surgery: A Randomized Clinical Trial. JAMA. 2020 Sep 1;324(9):848-858

Tharp WG, Murphy S, Breidenstein MW, Love C, Booms A, Rafferty MN, Friend AF, Perrapato S, Ahern TP, Dixon AE, Bates JHT, Bender SP: Body Habitus and Dynamic Surgical Conditions Independently Impair Pulmonary Mechanics during Robotic-assisted Laparoscopic Surgery. Anesthesiology. 2020 Oct 1;133(4):750-763

minor comments:

Introduction:

  1. Line 29: I don't think the reference 1 in the first sentence is appropriate and I think the study is overrated. I advise citing more appropriate literature, which should not be a problem.
  2. Line 30-31: I think the expression that mechanical forces generated in the ventilator lead to atelectasis misleading. It is mechanical ventilation with positive pressures and use of inadequate PEEP.Please change it, thank you.
  3. Line 31-34: Many studies are mentioned, but then only the 21-year-old study of the ARDS Network is cited. There are many more recent and better studies that should be considered and cited here.
  4. There is no mention of the fact that NO large RCT has yet been able to show evidence of a ventilation strategy that reduces PPC. So far all have been negative (ProVHILO, PROBESE). Why this is the case should be discussed, as it may be one reason why we need to look at ventilation in an individualised way and not "one size fits it all".

Materials and Methods:

  1. Line 53-56: Please define in more detail the selection criteria of the publications considered in the review. This is too general so far. Furthermore, it is interesting to know how many studies / publications were not considered and why? What is meant by high quality? Please elaborate.

The role of tidal volume (Are volutrauma and atelectrauma really unconnected?):

  1. Line 58: Why 6-7 ml/kg? Current recommendations repeatedly state 6-8 ml/kg. Please elaborate on this point, thank you.
  2. Line 58. Please add that ideal body weight is necessary as a basis for calculating the Vt.

How to set PEEP:

  1. Line 114-115: Different situations and patient groups are mentioned and will be discussed in more detail later. In particular, obese patients are a special group whose incidence continues to increase and yet have many respiratory physiological differences. Therefore, this group should be discussed in the manuscript when it comes to individualisation of ventilation.
  2. It should be made clearer in this paragraph why driving pressure was chosen as the global parameter.
  3. It is not clear to me in the manuscript on what scientific basis / evaluation Figure 1 is based. It seems to me that this is a new proposal by the authors, but it has not (yet) been tested for feasibility in a clinical setting and it is not known whether this enables an individualised PEEP setting that reduces the risk of PPC. This definitely needs to be added so that the reader perceives this only as a suggestion and can critically evaluate it.

A single variable to evaluate all the energy load: the mechanical power:

  1. In general, the formula for estimating the mechanical power is interesting. However, alternatives should be discussed on how to approach this topic. Ultimately, the scientific data and evaluation is rather thin, with only one study of 80 patients. A review of this formula in a larger patient population, which is also clearly more heterogeneous, is still pending.
  2. Line 135: What is the 12 J/min based on? This must be stated.
  3. Line 136: In my opinion, the study mentioned as citation 30 was not designed to evaluate this aspect and another citation should be given. If this is not possible, the paragraph should be critically revised.
  4. In general, the formula is only a global estimate and regional differences cannot be taken into account. Despite low values of the formula, local damage is possible and even probable if inhomogeneous ventilation with atelectases on one side and hyperinflation on the other side are present at the same time. This should be critically discussed in order to critically assess the validity of the formula.
  5. Line 143-147: It has not yet been shown in humans that higher PEEP automatically increases mechanical performance. Therefore, the conclusion cannot be drawn and is not an argument for less PEEP. In this respect, the paragraph needs to be changed.

Special settings: laparoscopy surgery:

  1. Figure 2 is a suggestion by the authors and has not been evaluated by clinical trials. This must be made clear in the explanation.

Special settings: Thoracic surgery

  1. Some literature citations do not seem to fit the statements and should be critically revised. This concerns: line 285 ref. 75, line 288 ref. 63 and line 290 ref. 66
  2. Figure 3: It should be made clear that this is a proposal for intraoperative ventilation adjustment that has not been evaluated in clinical trials. The name is also misleading as we do not know if this pathway is truly protective.

From protective to personalized: the future of intraoperative mechanical ventilation

  1. Lines 321-325: The aspect of EIT and the possibilities arising from the technology should be discussed more. There are already other studies on EIT that make it possible to individualise PEEP on the basis of additional information. These should be included in the discussion (Nestler et al. BJA 2017 or Girrbach et al. BJA 2020).
  2. Line 322. There is a typo concerning “weather” (should be “whether”)